# Prune or quantize? Strategy for Pareto-optimally low-cost and accurate CNN

## Abstract

Pruning and quantization are typical approaches to reduce the computational cost of convolutional neural network (CNN) inference. Although the idea of combining both approaches seems natural, it is surprisingly difficult to determine the effects of the combination without measuring performance on the specific hardware that the user will use. This is because the benefits of pruning and quantization strongly depend on the hardware architecture where the model is executed. For example, a CPU-like architecture with no parallelization may fully exploit the reduction of computations by unstructured pruning to improve speed, but a GPU-like massive parallel architecture would not. Further, there have been emerging proposals of novel hardware architectures, such as those supporting variable bit-precision quantization. From an engineering viewpoint, optimization for each hardware architecture is useful and important in practice, but this is in essence a brute-force approach. Therefore, in this paper, we first propose a hardware-agnostic metric for measuring computational costs. Using the proposed metric, we demonstrate that Pareto-optimal performance, where the best accuracy is obtained at a given computational cost, is achieved when a slim model with fewer parameters is moderately quantized rather than a fat model with a huge number of parameters is quantized to extremely low bit precision, such as binary or ternary. Furthermore, we empirically find a possible quantitative relation between the proposed metric and the signal-to-noise ratio during stochastic gradient descent (SGD) training, by which information obtained during SGD training provides an optimal policy for quantization and pruning. We show the Pareto frontier is improved by $4\times$ in a post-training quantization scenario based on these findings. These findings not only improve the Pareto frontier for accuracy versus computational cost, but also provide new insights into deep neural networks.

## 1 Introduction

Reducing execution cost of deep learning inference is one of the most active research topics for applying superhuman recognition in embedded IoT devices and robots. A typical approach for employing memory- and computation-efficient components is separable convolution, which is a combination of depth-wise and point-wise convolutions (Iandola et al., 2016; Zoph et al., 2018; Zhang et al., 2018; Howard et al., 2017), structured/unstructured pruning of connections and activations, and quantizing activation, weight, and their vectors (Stock et al., 2019; Jegou et al., 2011; Gong et al., 2014). Among these, separable convolution and structured pruning are similar, in that separable convolution can be viewed as convolutions pruned in a handcrafted manner. From a pruning viewpoint, since the separable convolution structure results from applying aggressive pruning to normal convolution, the result is drastic reductions in memory and computational cost at the expense of greatly decreased accuracy (Stock et al., 2019). On the other hand, structured pruning and quantization are seemingly orthogonal approaches that can be naturally combined (Tung & Mori, 2018; Han et al., 2016). However, their interactions are still not well-studied. For instance, the use of a single-bit representation is being actively explored as an extreme quantization. Since a nonnegligible accuracy drop is inevitable in extreme quantization, some papers have proposed increasing the number of channels to compensate for the lack of expressivity (Lin et al., 2017). In other words, a quantization approach can further reduce the number of bits by compromising the increase in number of channels, or the increase in number of computations. This indicates that, con-

versely, reducing channels by pruning may limit capability for quantization. This discussion raises a controversial question: which is better, a fat model with smaller bit width or a slim model with larger bit width? Answering this question requires a metric that fairly measures the effects of both pruning and quantization. One such metric in the literature is the inference speed when the model is executed on specific hardware. This metric is useful or even ideal when the target hardware is known in advance but strongly depends on features of the hardware architecture. Yang et al. (2018) searched for an optimal architecture using inference time as the optimization objective and found different optimal architectures depending on the target device. For example, if the hardware cannot handle extremely low bit-widths (1 or 2 bits), instead treating them as 8-bit integers with upper bits filled with zeros, we cannot exploit the reduction of bit width to improve inference speed. From a theoretical viewpoint, figuring out the extent to which we can reduce the computational complexity of deep neural networks is another important open question.

The discussion so far urges us to develop a hardware-agnostic and theoretically reasonable metric for measuring computational costs of neural network architectures. In this paper, we propose the Frobenius norm of the effective value of weight parameters as one such metric. This metric is proportional to the total energy when the model is executed on *ideal* hardware, where energy consumption for a single multiply-accumulate (MAC) computation is proportional to the squared effective amplitude of the individual weight parameter used for the MAC computation. The basic idea of the metric is analogous to a highly efficient class-B amplifier circuit whose energy consumption is determined by the instant signal amplitude (Sechi, 1976). This metric successfully reflects the effects of both quantization and structured/unstructured pruning in accordance with intuition.

Using the proposed metric, we empirically find that a slimmer model can achieve a far better Pareto frontier in a lower computational cost region than can a fatter model after quantization, while a fat model is advantageous for achieving higher accuracy in a larger computational cost region. Finally, we perform experiments under a post-training quantization scenario (Banner et al., 2018) on ImageNet dataset (Deng et al., 2009) to verify the validity of our claim, namely that prune-then-quantize is superior to quantize-only or prune-only for achieving a better Pareto frontier.

Further, since this metric is relevant to the signal-to-noise ratio ($S/N$), it is measurable during SGD training, in which the absolute value of weights and the random walk of weight parameters correspond to signal and noise, respectively. We observe that the dependencies of the metric on validation accuracy seem to be correlated between those during training and those applying quantization after training. From this observation, we point out some possibilities for which we could expect robustness of a model for quantization from information obtained during training, we could determine an optimal policy for quantization of that model, and we could develop a novel optimization or regularization scheme.

The main contributions of this paper are as follows:

- We define a hardware-agnostic metric for measuring the computational cost of pruned and quantized models.

- We empirically find that models with fewer parameters achieve far better accuracy in a low computational cost region after quantization.

- We show a potential quantitative relation between quantization noise and perturbation of weight parameters during SGD training.

And as implications, we hope to exploit our findings for

- thorough comparison of various neural network architectures using the proposed hardware-agnostic metric,

- development of a method for extracting a quantization policy from information obtained during SGD training, and

- development of a training algorithm or regularization scheme for producing robust models based on the relation between quantization noise and perturbation of weight parameters during SGD training.

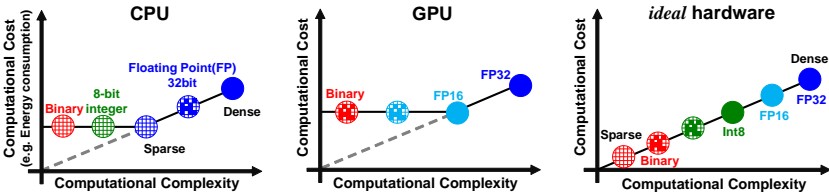

Figure 1: Left/Center: Computational cost strongly depends on the hardware architecture on which the model is executed. Right: Proposed computational cost for analysis or theoretical research, assuming an *ideal* hardware architecture.

## 2 EFFECTIVE SIGNAL NORM

We seek a metric that properly reflects the effects of both quantization and pruning. Conventionally, quantization effectiveness is evaluated according to the number of bits required to achieve a given accuracy, or the accuracy achieved by using certain bit numbers for specific network architectures (Stock et al., 2019). We cannot use this to compare efficiencies between different architecture models (e.g., MobileNet versus ResNet-18). The number of MAC computations or parameters can be used to compare different architectures, but the number of MAC computations does not consider quantization and the number of parameters is not directly related to inference time.

Recently, the use of actual or estimated inference speeds as a metric for comparing network architectures has been proposed (Yang et al., 2018; Wang et al., 2019a; Cai et al., 2019). This metric is very useful when the target hardware is known in advance, and ideal for those who wish to use the model that performs best on that hardware. However, this metric is strongly hardware dependent. Indeed, Yang et al. (2018); Wang et al. (2019a); Cai et al. (2019) found that optimal architectures for different types of target hardware are totally different. Considering interest in, for example, the simplest realizable deep neural network model while achieving a required accuracy, there is a need for a hardware-agnostic metric.

The metric for model evaluation should correlate with energy consumed when the model is executed on *ideal* hardware. We assume that energy consumption by *ideal* hardware monotonically decreases when the bit width is reduced by quantization and when the number of nonzeros in weight parameters is reduced by pruning. For example, hardware with an 8-bit integer MAC array cannot be further accelerated even if the bit width is reduced from 8 to 1 or 2 bits. Thus, the energy consumption measured using such hardware does not satisfy the aforementioned requirement and cannot be our metric. Hardware like a CPU, which processes each computation in serial, can naturally exploit the structured or unstructured sparsity of weight parameters by skipping computations with zeroed weights. However, because it is difficult to parallelize computations while maintaining such a strategy, it is generally difficult to benefit from sparsity in GPU-like hardware employing massively parallel MAC units. Hardware dedicated to sparse convolution (Lu et al., 2019) tends to show better performance only when sparsity is sufficiently high, due to relatively large overheads for encoding and decoding sparse weight parameters in a special format.

Therefore, the benefit of sparsity from pruning and low bit width from quantization largely depends on the hardware architecture, so long as we consider only existing hardware. Because we require a hardware-agnostic metric, we assume *ideal* hardware in which energy consumption is linearly proportional to the number of nonzero weight parameters and monotonically depends on the bit width of weight parameters, as shown in Figure 1, setting aside the feasibility of such *ideal* hardware.

### 2.1 DEFINITION OF EFFECTIVE SIGNAL NORM

We define a metric called the *effective signal norm* (ESN) as

$$\text{ESN} = \sum_l ||c^l f(\mathbf{W}_{\text{int}}^l)||_F^2, \tag{1}$$

with $\mathbf{W}_{\text{int}}^l = \lfloor \mathbf{W}^l / \Delta^l \rfloor + 0.5$, where $\mathbf{W}^l$ is the weight tensor and $\Delta^l$ is the quantization step size of the $l$th layer; and $c^l$ is a coefficient depending on the layer, in that if $c^l = 1$, ESN is related to the number of parameters (cf. memory footprint), and if $c^l$ is the number of computations per parameter at the $l$th layer, ESN is related to the number of computations (cf. FLOP). $f(\cdot)$ is an element-wise function that determines how the metric responds to the value of each weight parameter. We propose two functions for $f(\cdot)$. The first is $f(\mathbf{W}_{\text{int}}^l) = \mathbf{W}_{\text{int}}^l$, based on the assumption that energy consumption increases with the square of the value for each weight parameter or for each computation. When $c^l = 1$, the definition is

$$\text{ESN}_a = \sum_l ||\mathbf{W}_{\text{int}}^l||_F^2. \tag{2}$$

This assumption is reasonable when we employ an analog (or in-memory) MAC computation engine (Shafiee et al., 2016; Miyashita et al., 2017), because energy consumption is proportional to the square of the signal amplitude when the signal represents an analog quantity such as voltage or current. Assuming *ideal* hardware, we adopt a definition where energy consumption varies according to the instant amplitude (cf. class-B amplifier), which is more energy efficient than the case where energy consumption is constant and the value is determined by the maximal amplitude (cf. class-A amplifier) (Sechi, 1976).

The second proposed function is $f(\mathbf{W}_{\text{int}}^l) = \lceil \log_2(\text{abs}(\mathbf{W}_{\text{int}}^l)) + 1 \rceil$, where $\log_2(\cdot)$ and $\text{abs}(\cdot)$ are functions applied to each element of a tensor argument. This is based on the assumption that energy consumption increases with the binary logarithm of the value for each weight parameter. When $c^l = 1$, the definition is

$$\text{ESN}_d = \sum_l ||\lceil \log_2(\text{abs}(\mathbf{W}_{\text{int}}^l)) + 1 \rceil||_F^2. \tag{3}$$

In a digital circuit, a number is represented as a binary digit (bits), so the energy consumption for moving or processing signals is roughly proportional to the number of bits, which is the binary logarithm of the value. It is therefore reasonable to use Equation (3) for a digital circuit.

## 2.2 RELATION BETWEEN ESN AND S/N

The effective signal norm defined in Equation (2) is related to the signal-to-noise ratio ($S/N$) when quantization noise is dominant and noise is approximated by a uniform distribution (Gray & Neuhoff, 1998).

$$\text{ESN}_a = \sum_l \left( \frac{||\mathbf{W}^l||_F^2}{||\mathbf{W}^l - \Delta \cdot \mathbf{W}_{\text{int}}^l||_F^2} \cdot \frac{||\mathbf{W}^l||_0}{12} \right) = \sum_l \left( S/N_l \cdot \frac{||\mathbf{W}^l||_0}{12} \right), \tag{4}$$

where $l$ is the layer index, $S/N_l$ is the signal-to-quantization-noise ratio of the $l$th layer as defined by $S/N_l = \frac{||\mathbf{W}^l||_F^2}{||\mathbf{W}^l - \Delta \cdot \mathbf{W}_{\text{int}}^l||_F^2}$, and $||\mathbf{W}^l||_0$ is the number of nonzero elements in the tensor $\mathbf{W}^l$. Appendix E presents the derivation of Equation (4). This equation allows us to calculate $\text{ESN}_a$, so long as $S/N$ is defined.

## 2.3 $\text{ESN}_a$ DURING TRAINING

For example, we can define $S/N$ by regarding perturbation of weight parameters during training as noise. Formally, we define the signal and noise at the $j$th epoch as

$$S_j = \sum_l \sum_i ||\mathbf{W}_{j,i}^l||_F^2, \quad N_j = \sum_l \sum_i ||\mathbf{W}_{j,i}^l - \mathbf{W}_{\text{init}_j}^l||_F^2, \tag{5}$$

where $\mathbf{W}_{j,i}^l$ is the weight parameters in the $l$th layer at the $i$th iteration in the $j$th epoch, and $\mathbf{W}_{\text{init}_j}^l$ is a snapshot of the weight at the beginning of the $j$th epoch. Then, $N_j$ is the effective value of random walk noise for weight parameters in one epoch.

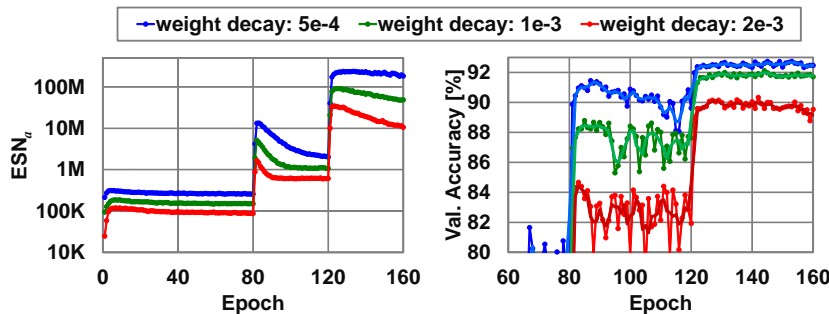

Figure 2: Training curves of $\text{ESN}_a$ (left) and validation accuracy (right) on CIFAR-10. In the right graph, moving average curve between 3-epochs is overlapped on each plot.

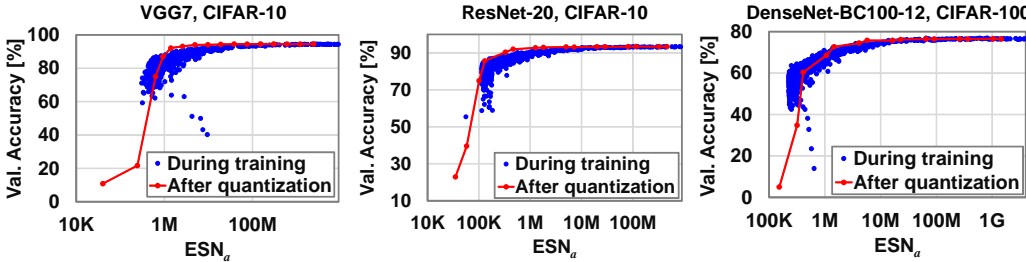

Figure 3: Relation between validation accuracy and $\text{ESN}_a$ during training and after quantization. $\text{ESN}_a$ of the former is computed using Equation (5), and that of the latter is computed using Equation (2)

The left figure in Figure 2 shows the $\text{ESN}_a$ curve during training, which is calculated by Equation (4), and $S/N$ as defined by Equation (5). In this experiment, we use ResNet-20 (He et al., 2016) on CIFAR-10 dataset (Krizhevsky, 2009). We use SGD with momentum 0.9, and we set the mini-batch size to 100 and the initial learning rate to 0.1, which is shifted by $1/10$ at epochs 80 and 120. We vary the weight decay factor from $5 \times 10^{-4}$ to $2 \times 10^{-3}$, indicated by different colors in Figure 2.

Interestingly, the $\text{ESN}_a$ curves look very similar to the validation accuracy curves shown in the right figure in Figure 2. Both $\text{ESN}_a$ and validation accuracy steeply increase at the point where the learning rate is decreased by $1/10$, because decreasing the learning rate reduces perturbation (or random walk) of weight values during one epoch, increasing $S/N$. We can also see that as the weight decay factor increases, $\text{ESN}_a$ tends to decrease. This is also reasonable, because weight decay decreases $||\mathbf{W}||_F^2$ or the signal level, reducing $S/N$. More interestingly, even this trivial tendency appears correlated with validation accuracy; for example, after a steep rise at epoch 80, both $\text{ESN}_a$ and validation accuracy gradually decrease, converging to stable values after overshooting. There is a similar relation between the amount of overshoot and weight decay factor, namely, we observed a larger overshoot with a smaller weight decay factor. These findings might be useful for developing a novel optimization algorithm, but we leave this for future work.

The blue plots in Figure 3 show $\text{ESN}_a$ versus validation accuracy. In this experiment, we use VGG7 (Simonyan & Zisserman, 2014) and ResNet-20 on CIFAR-10, and DenseNet-BC with $l = 100$, $k = 12$ (Huang et al., 2017) on CIFAR-100. We employ SGD with momentum 0.9 and set the initial learning rate to 0.1, followed by cosine annealing without restart (Ilya Loshchilov, 2017). The mini-batch size and weight decay factor are respectively set to 125 and $5 \times 10^{-4}$.

The red curve shows the $\text{ESN}_a$ versus validation accuracy curve when applying quantization to weight parameters trained by the training process where the blue plots are obtained. As the figure shows, the red and blue curves seem to be correlated. This suggests that the model acquires robustness to quantization by having experienced similar perturbations due to a random walk during the

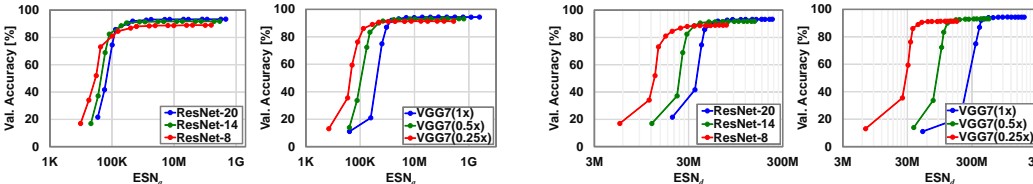

Figure 4: Accuracy versus $\text{ESN}_a$ (left) and $\text{ESN}_d$ (right) for various network depths and widths.

training process. If so, the $\text{ESN}_a$ versus validation accuracy curve during training is available as a loose boundary on accuracy degradation due to decrease of $\text{ESN}_a$ by quantization. We discuss this hypothesis in Section 2.7 again.

## 2.4 ESN VERSUS ACCURACY FOR VARIOUS MODEL SIZES

With our metric defined, we next attempt to investigate which is better, a model that is slim (with the number of weight parameters and computations reduced by pruning) and mildly quantized (e.g., with 4–5 bits), or a model that is fat (with a large number of weight parameters and computations) and radically quantized (e.g., with 1–2 bits). We evaluate six network architectures with different depths and widths: ResNet-8, 14, 20 and VGG7 with different channel widths ($1\times$, $0.5\times$, $0.25\times$) on CIFAR-10 dataset. In this experiment, we employ networks with originally small number of parameters as slim models instead of pruning fat models. The network architecture of ResNet is based on the original one without bottleneck architecture (He et al., 2016). The detailed network architecture of VGG7 is shown in Appendix F.

Figure 4 shows validation accuracy versus $\text{ESN}_a$ and $\text{ESN}_d$. Here, we consider differences in $\text{ESN}_a$ and $\text{ESN}_d$. For simplicity, we assume that all weight parameters have the same value $v > 1$, and that the number of weight parameters is $N$. The definitions of $\text{ESN}_a$ and $\text{ESN}_d$ are expressed as follows,

$$\text{ESN}_a = v^2 \times N, \quad \text{ESN}_d = \lceil \log_2 v + 1 \rceil \times N. \tag{6}$$

Therefore, when we compare $\text{ESN}_a$ and $\text{ESN}_d$, both are linearly proportional to the number of weight parameters, whereas values of weight parameters affect the metric at squared and logarithmic scales, respectively. This means that $\text{ESN}_a$ is more strongly dependent on the size of the values than on the number of weight parameters, and the opposite is true for $\text{ESN}_d$. In other words, $\text{ESN}_d$ is more sensitive to pruning than to quantization, and $\text{ESN}_a$ is more sensitive to quantization than to pruning. Figure 4 clearly shows these tendencies. We can see that even when using the $\text{ESN}_a$ metric, which is advantageous for quantization, the slimmer model shows better validation accuracy in lower ESN regions. These experiments indicate that in lower ESN regions, or with lower energy consumption by *ideal* hardware, especially when employing digital computing, it is an advisable strategy to prune the model to the limit where the desired accuracy is achieved, and then to apply quantization to obtain the highest possible accuracy. In contrast, it is a bad strategy to apply quantization to a fat model, even if it shows much better accuracy than the pruned slim model before quantization. Consequently, to the question posed at the beginning of this subsection—whether a slim and mildly quantized model or a fat and radically quantized model is better—the answer is the former one, which may in a sense raise a question on trends in quantization research. Note that the above discussion also indicates that the importance of quantization will be relatively increased if analog computing become practical in the future.

## 2.5 TRAINING UNDER LOW-ESN CONDITIONS

We next observe how the weight parameters evolve in the training process. Figure 5 shows from left to right the loss, $\text{ESN}_a$, and number of pruned filters (output channels). In this experiment, we train ResNet-20 on CIFAR-10 dataset with initial weight parameters of a trained model. We employ SGD with momentum 0.9, weight decay factor $5 \times 10^{-4}$, and mini-batch size 200, updating for 20 epochs. The learning rate (LR) is constant, with a value varied from 0.32 to 1.81 for each training. Note that we do not intentionally prune filters in this experiment; rather, filter pruning spontaneously occurs along with sequential updates of weight parameters by SGD. The center graph shows that as

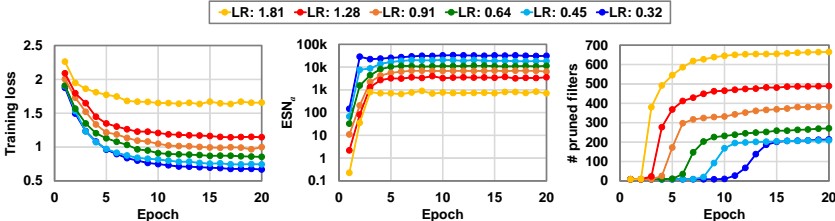

Figure 5: Training curves for loss, $\mathrm{ESN}_a$, and number of pruned filters under different learning rate (LR) conditions. $\mathrm{ESN}_a$ is calculated by Equations (4) and (5).

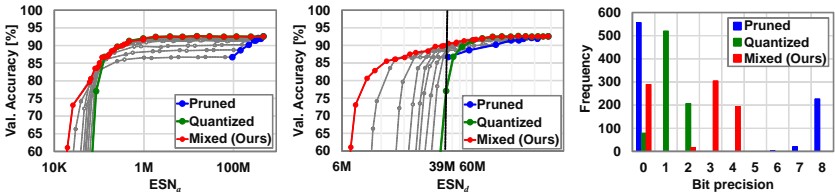

Figure 6: Accuracy versus $\mathrm{ESN}_{a/d}$ (left/center) and histogram of the maximal bit width for each filter (right) in the pruning only, quantization only, and prune-then-quantize mixed cases.

the learning rate increases, $\mathrm{ESN}_a$ decreases due to the larger noise. The number of pruned filters after epoch 20 is larger with the larger learning rate, which corresponds to smaller $\mathrm{ESN}_a$. In the training process, rather than continuing to increase at a constant rate throughout the training process, the number of pruned filters steeply increases within a few epochs and then is maintained without noticeable change.

This indicates that under a small $\mathrm{ESN}_a$ condition controlled by the learning rate, the optimizer finds a solution for reducing the number of parameters by pruning filters and prioritizing increases in amplitude of the remaining parameters, rather than letting all parameters survive with small amplitude.

## 2.6 PRUNE *then* QUANTIZE

In the previous two subsections, we empirically showed that achieving optimal accuracy in a low ESN region requires preparation of a slim model before applying quantization. A slim model can be generated by pruning a fat model or by designing a slim architecture. Based on the lottery ticket hypothesis (Frankle & Carbin, 2019), where a fat or overparameterized network can have higher potential for finding better optimal parameters by SGD training, we adopt the former method, namely, pruning weight parameters from a pretrained fat model. We apply ADMM regularization for structured pruning (Wang et al., 2019b), since this achieves state-of-the-art performance.

In this experiment, we evaluate ResNet-20 on CIFAR-10 dataset. We update pretrained weight parameters through the SGD algorithm with alternating direction method of multipliers (ADMM) regularization (Wang et al., 2019b) for a few additional epochs. We then fine-tune the resulting pruned model over 160 epochs. In this fine tuning, we employ SGD with momentum 0.9 and set the mini-batch size to 100 and the initial learning rate to 0.1, followed by cosine annealing without restart. The weight decay factor is set to $5 \times 10^{-4}$. Since the weight parameters are not yet quantized in this phase, we update all parameters, except that we set weights for the filters (output channels) and channels (input channels) pruned in the previous phase to zero. Finally, we apply quantization.

As Figure 6 shows, the Pareto frontier significantly improves by applying prune-then-quantize (red), as compared to applying extreme quantization (green). The right graph in Figure 6 shows a histogram of the maximal bit width under each filter (output channel) of the three models indicated by the broken line in the center graph ($\mathrm{ESN}_d = 39\mathrm{M(bits)}$). Since the pretrained model has 800 filters before pruning and quantization, summations of frequencies for each color are 800. In the pruning only (blue) case, the weight parameters in 555 filters, the largest number among the three methods,

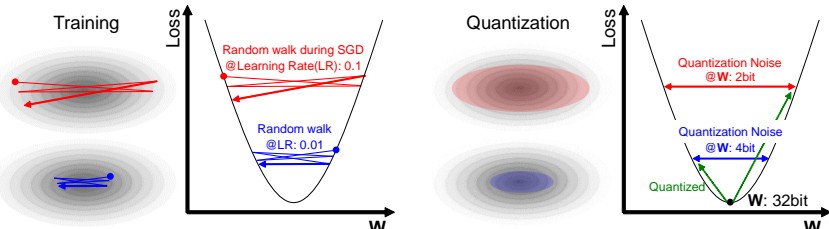

Figure 7: Relation between random walk during SGD and quantization noise. Loss often steeply decreases when LR is shifted by (e.g., $1/10$). In this situation, loss before shifting LR is possibly governed by random walk noise due to the large LR, as shown in the left figure. When we quantize weight parameters of the trained model, however, the loss or accuracy should be determined by noise induced by quantization, as shown in the right figure. If we can assume that the loss landscapes of the two cases should not diverge too much, the loss (or accuracy) after quantization is possibly predicted by the dependency of loss (or accuracy) on $\text{ESN}_a$ during training.

are pruned (represented as $0$ bits). Most of remaining filters are $8$ bits. In the quantization only (green) case, the number of pruned filters is a low $77$, while the bit width of most remaining filters is $1$ bit. In the proposed prune-then-quantize mixed (red) case, the number of pruned filters is $288$ and the bit widths of most remaining filters are $3$ or $4$ bits, so the remaining filters are mildly quantized. As this case shows, while the $\text{ESN}_d$ values are similar, the validation accuracies differ depending on the number and bit width of the weight parameters. The prune-then-quantize mixed method properly produces a pruned and mildly quantized model, which achieves far better validation accuracy, especially in the low ESN region, as compared to the model produced by extremely quantizing a fat model without pruning.

## 2.7 $\text{ESN}_a$ FOR QUANTIZATION

As Section 2.3 showed, there is a possible quantitative relation between $\text{ESN}_a$ during model training and $\text{ESN}_a$ of the pruned and quantized version of that model after training. This finding inspires us to exploit $\text{ESN}_a$ information obtained during training to determine the quantization policy, for example, how many bits to allocate for each layer. Figure 7 shows the intuition behind this idea, namely straightforward use of the $\text{ESN}_a$ values for each layer at a certain epoch during training as targets for quantization. Appendix B describes a preliminary experiment for verifying this idea.

## 3 EXPERIMENT ON IMAGENET

To verify the validity of our claims, we performed experiments under a post-training quantization scenario (Banner et al., 2018) on ImageNet dataset (Deng et al., 2009). Post-training quantization assumes a use-case scenario where a user (e.g., at an edge side) quantizes a given pretrained CNN model with a limited number of unlabeled training samples for obtaining a lightweight model suited to a particular situation. As concluded in Section 2.4, in a low ESN region or with low energy consumption, we can achieve higher accuracy by quantizing a slim model than by quantizing a fat model, even if the latter is more accurate. To provide optimal accuracy in a low computational-cost region, we apply the proposed prune-then-quantize method to the post-training quantization scenario.

Figure 8 shows the result of applying our method to ResNet-18 and ResNet-50 on ImageNet dataset. We used pretrained models[1][2] as basic models with $1\times$ channel width, and prune the models to various channel widths ($0.25\times$, $0.5\times$, and $0.75\times$). In this experiment, we determine which filters (output channels) to prune based on the signal norm of weights calculated by Equation (5). We prune filters in order from the smallest signal norm, and tune each model for the pruned structure through the SGD algorithm over $4$ epochs. The mini-batch size is set to $64$, and the learning rate is initialized

---

[1]https://download.pytorch.org/models/resnet18-5c106cde.pth,
[2]https://download.pytorch.org/models/resnet50-19c8e357.pth

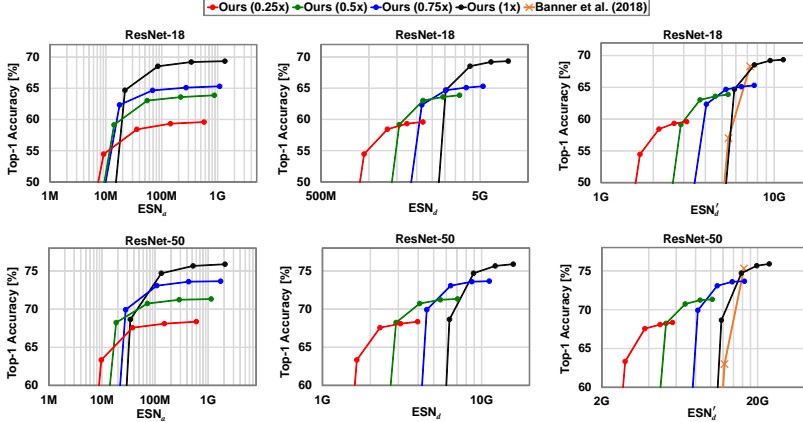

Figure 8: (Left/center) top-1 accuracy on ImageNet versus $\text{ESN}_{a/d}$ and (right) estimated $\text{ESN}_d'$, applying the prune-then-quantize method to (upper) ResNet-18 and (lower) ResNet-50.

to $0.1$ and divided by $10$ at epochs $2$ and $3$. We then quantize each model. For quantization, we only fine-tune statistic parameters in BatchNorm layers (called `running_mean` and `running_var` in pytorch (Paszke et al., 2017)) with no labeled data (Sasaki et al., 2019), which is the same condition as conventional work (Banner et al., 2018).

In addition to evaluations by $\text{ESN}_a$ and $\text{ESN}_d$, to compare our method with the conventional work, we estimate its computational cost using Equation (1) with $c^l$ and $f(\cdot)$ properly defined, as

$$\text{ESN}_d' = \sum_l \sum_m \text{MAC}_{l,m} \times \lceil \log_2(\max(\text{abs}(\mathbf{W}_{\text{int}}^{l,m}))) + 1 \rceil, \tag{7}$$

where $\text{MAC}_{l,m}$ is the number of computation per filter and $\mathbf{W}_{\text{int}}^{l,m}$ is the $m$th filter of $\mathbf{W}_{\text{int}}^l$. By applying the $\max(\cdot)$ function, we restrict the number of bits to be same in each filter.

We apply this metric[3] to a recently proposed efficient MAC array architecture capable of handling variable bit precision (Maki et al., 2018).

As Figure 8 shows, significantly using slim models after pruning some filters improves the Pareto frontier as compared to Banner et al. (2018). For example, in ResNet-18, $57\%$ top-1 accuracy is achieved at $2\times$ lower computational cost, and in ResNet-50, $63\%$ top-1 accuracy is at $4\times$ lower computational cost. These results suggest that we should adopt a slim model to achieve best accuracy in low computational cost regions, instead of quantizing weight parameters to an extremely low bit precision.

## 4 CONCLUSION

We proposed a hardware-agnostic metric called the effective signal norm (ESN) to measure computational costs. Using this metric, we demonstrated that a slim model with fewer weight parameters achieves better Pareto frontier performance in low computational cost regions than does an extremely quantized fat model. We also showed a possible quantitative relation between weight perturbation during SGD training and quantization noise or robustness against quantization.

By defining this metric, on the hardware architecture side we can aim at realizing hardware whose energy consumption is proportional to the metric. On the algorithmic side, we can reduce the metric. We therefore expect consensus-sharing regarding the metric for computational cost to accelerate research progress for both algorithms and hardware architectures.

---

[3]In Maki et al. (2018), this metric is defined as "MAC×bit", which is essentially same as $\text{ESN}_d'$.

## 5 RELATED WORKS

**Quantization** Courbariaux & Bengio (2016); Rastegari et al. (2016); Zhu et al. (2017) quantize weight parameters and activations as 1 or 2. Since such extreme quantization deteriorates accuracy to a nonnegligible extent, quantization methods with, for example, 4–8 bits are also actively explored (Banner et al., 2018; Lin et al., 2017) to avoid the accuracy drop. Miyashita et al. (2016) attempts to quantize weight parameters and activations in a logarithmic domain, aiming not only at reducing information loss but also replacing multiplication with bit-shift to simplify computation. To reduce the *average* bit width as much as possible while maintaining accuracy, and to exploit it to accelerate inference speed, Maki et al. (2018) proposes variable bit-width quantization co-optimized with hardware architecture. Although nonuniform quantization (Tung & Mori, 2018; Han et al., 2016) and vector or product quantization (Gong et al., 2014; Jegou et al., 2011; Stock et al., 2019) are also actively studied, these approaches are effective for reducing the memory footprint but not for directly reducing computational costs. Use cases include quantization-aware training (Courbariaux & Bengio, 2016; Rastegari et al., 2016; Zhu et al., 2017; Lin et al., 2017; Zhang et al., 2018) and post-training quantization (Banner et al., 2018). Quantization-aware training achieves better accuracy with lower bit widths, and is almost inevitable for those with extreme quantization. However, it tends to incur higher training costs in most cases. The post-training quantization scenario, which is executable with less computational resources and a smaller training dataset, has potentially more applications, so we also target this scenario.

**Pruning** Pruning methods include structured pruning, which prunes whole layers, filters, or channels to maintain a regular structure so that computations are easily parallelized, and unstructured pruning, which randomly prunes individual weight parameters. Since it is difficult to benefit from unstructured pruning unless sparsity is sufficiently large, due to its incompatibility with parallelization (Lu et al., 2019), we target structured pruning. Many pruning methods have also been proposed, such as criteria-based approaches (LeCun et al., 1990), regularization-based approaches (Han et al., 2015; Wang et al., 2019b), and methods employing reinforcement learning (Zhong et al., 2018). In this paper, we argue that a pruned slim model performs better after quantization in lower computational cost regions. Any pruning method can be applied to produce a pruned slim model.

**Metric for measuring cost** In most quantization papers, the standard metric is accuracy achieved under a certain bit width. For pruning, the standard metric is the number of parameters or the number of computations (FLOP). Some papers use inference time when the model runs on specific hardware (Cai et al., 2019; Yang et al., 2018), but this metric is strongly hardware-dependent. In this paper, we propose a hardware-agnostic metric.

**Noise during training** Relations between learning rate, batch size, and noise during training are widely discussed (Keskar et al., 2017; Xing et al., 2018). To make the model more robust to quantization, some papers propose intentional addition of noise to gradient or weight parameters (Spallanzani et al., 2019; Baskin et al., 2018a;b).

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

## A    DETAILS OF THE QUANTIZATION PROCESS

We basically apply a midrise-type quantization (Gersho, 1977), but modify it as follows. In the quantization phase, we prune filters in which all weight parameters are within $\pm\Delta/2$, as described in the following pseudocode with Numpy (Oliphant, 2006) notation.

```
# Quantize
# W: (filter(out_ch), channel(in_ch), ky, kx)
pruned = numpy.all(abs(W) < delta/2, axis=(1,2,3))
W_int = numpy.floor(W/delta) + 0.5
W_int[pruned, :, :, :] = 0
```

## B    $\text{ESN}_a$ FOR OPTIMAL QUANTIZATION POLICY

We attempt to exploit the $\text{ESN}_a$ obtained during training to determine a quantization policy using VGG7 where numbers of filters in all convolutional layers are reduced to $0.25\times$ on CIFAR-10. The left figure in Figure 9 show the validation accuracy versus $\text{ESN}_a$ during training and after quantization. As an initial policy, we apply the same bit width for all layers. The results shown in gray do not fit the blue plot during training, possibly indicating the policy is suboptimal. We next investigate the $\text{ESN}_a$ of each layer and find that the misfit is caused by the sixth layer, as shown in the right figure. When we consider validation accuracy versus the sixth-layer $\text{ESN}_a$, shown in green symbols, validation accuracy after quantization deteriorates at a much higher $\text{ESN}_a$ than during training. An interpretation is that since this deterioration is caused by other layers (e.g., the first) rather than the sixth layer, weight parameters in the sixth layer should be more aggressively quantized. Based on this observation, we modify the quantization policy such that the bit width of the sixth layer become smaller than that in other layers, thereby improving performance as shown by the red plot in the left figure. This preliminary experimental result suggests we can use $\text{ESN}_a$ information during training to find an optimal quantization policy.

## C    RESNET-18 VERSUS RESNET-50

The proposed metric allows us to compare different network architectures. In Figure 10, we compare the Pareto frontier of validation accuracy versus $\text{ESN}_{a/d}$ curves for ResNet-18 and ResNet-50. The plots in Figure 10 are extracted from Figure 8. This result shows that ResNet-50 has a better Pareto frontier than does ResNet-18. One convincing reason is that ResNet-50 employs a parameter-efficient bottleneck structure, whereas ResNet-18 does not. In contrast, ResNet-18 shows better performance for $\text{ESN}_d < 2G$, partly due to the higher sensitivity of $\text{ESN}_d$ to the number of parameters. This suggests use of ResNet-18 when employing digital computing in this $\text{ESN}_d$ region. Such comparisons enabled by the proposed metric are useful when searching for or developing efficient structures.

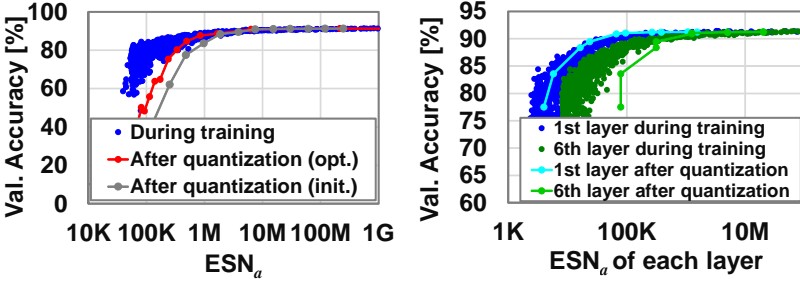

Figure 9: Relation between validation accuracy versus $\text{ESN}_a$ during training and after quantization.

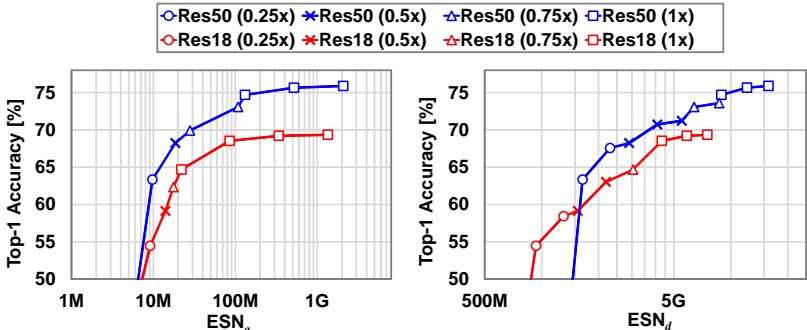

Figure 10: Comparison of ResNet-18 and ResNet-50 with respect to top-1 accuracy on ImageNet versus $\text{ESN}_{a/d}$ (left/right).

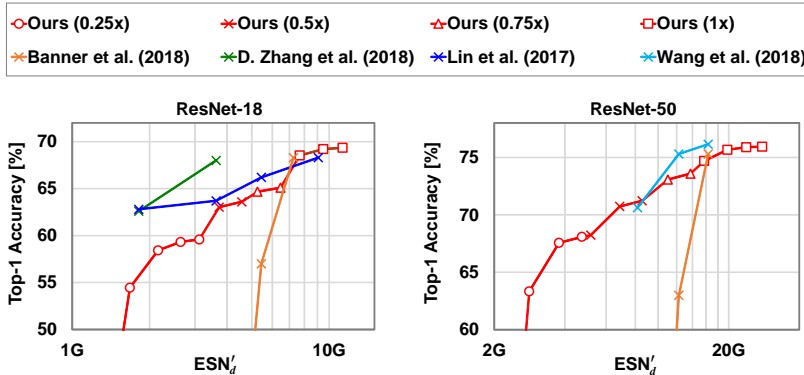

Figure 11: Comparison of other methods with respect to top-1 accuracy on ImageNet versus estimated $\text{ESN}'_d$.

## D    COMPARISON WITH OTHER METHODS

In Figure 11, we superimpose results from other works on Figure 8. Lin et al. (2017) and Zhang et al. (2018) show better accuracy in smaller $\text{ESN}'_d$ regions than ours, but these apply quantization-aware training, whereas our result is obtained by post-training quantization. Although we expect that the findings presented in this paper will also improve performance of quantization-aware training, we leave an explicit demonstration of this to future works.

## E    DERIVATION OF EQUATION (4)

The mean squared noise energy $N$ due to quantization over the all elements in a tensor $\mathbf{W}$ is computed as

$$N = ||\mathbf{W} - \Delta \cdot \mathbf{W}_{\text{int}}||_F^2 / ||\mathbf{W}||_0, \tag{8}$$

where $\mathbf{W}_{\text{int}} = \lfloor \mathbf{W}/\Delta \rfloor + 0.5$, $\Delta$ is the quantization step size and $||\mathbf{W}||_0$ is the number of nonzero elements in $\mathbf{W}$. When quantization noise is approximated by a uniform distribution, the mean squared noise energy is (Gray & Neuhoff, 1998)

$$N = \int_{-\Delta/2}^{\Delta/2} \frac{1}{\Delta} n^2 dn = \frac{\Delta^2}{12}. \tag{9}$$

From Equations (8) and (9),

$$\Delta^2 = ||\mathbf{W} - \Delta \cdot \mathbf{W}_{\text{int}}||_F^2 \times \frac{12}{||\mathbf{W}||_0}. \tag{10}$$

Then,

$$||\mathbf{W}_{\text{int}}||_F^2 \sim ||\mathbf{W}/\Delta||_F^2 = \frac{||\mathbf{W}||_F^2}{\Delta^2} = \frac{||\mathbf{W}||_F^2}{||\mathbf{W} - \Delta \cdot \mathbf{W}_{\text{int}}||_F^2} \cdot \frac{||\mathbf{W}||_0}{12}. \tag{11}$$

Here we again assume that quantization noise is uniformly distributed. In this equation, since $||\mathbf{W}||$ and $||\mathbf{W} - \Delta \cdot \mathbf{W}||_F^2$ are respectively the signal and noise norm, the term $\frac{||\mathbf{W}||_F^2}{||\mathbf{W} - \Delta \cdot \mathbf{W}_{\text{int}}||_F^2}$ is considered to be $S/N$, and thus,

$$||\mathbf{W}_{\text{int}}||_F^2 = \frac{||\mathbf{W}||_F^2}{||\mathbf{W} - \Delta \cdot \mathbf{W}_{\text{int}}||_F^2} \cdot \frac{||\mathbf{W}||_0}{12} = S/N \cdot \frac{||\mathbf{W}||_0}{12}. \tag{12}$$

We obtain Equation (4) by summing this value over all layers.

## F  DETAILS OF NETWORK ARCHITECTURES

The network architecures of VGG7 ($1\times$, $0.5\times$, $0.25\times$) which we use in section 2.4 are shown in table 1. These architectures are based on the original model of VGG (Simonyan & Zisserman, 2014).

Table 1: Detailed network architecture of VGG7 for CIFAR-10. $C_I$ and $C_O$ represent the number of input and output channels, respectively in conv-layer. In fc-layer, they represent the number of input and output neurons

| Layer name | VGG ($1\times$) | | VGG ($0.5\times$) | | VGG ($0.25\times$) | | kernel size | padding | stride |
|---|---|---|---|---|---|---|---|---|---|
| | $C_I$ | $C_O$ | $C_I$ | $C_O$ | $C_I$ | $C_O$ | | | |
| conv1 | 3 | 128 | 3 | 64 | 3 | 32 | $3 \times 3$ | 1 | 1 |
| conv2 | 128 | 128 | 64 | 64 | 32 | 32 | $3 \times 3$ | 1 | 1 |
| maxpool | 128 | 128 | 64 | 64 | 32 | 32 | $2 \times 2$ | 0 | 2 |
| conv3 | 256 | 256 | 128 | 128 | 64 | 64 | $3 \times 3$ | 1 | 1 |
| conv4 | 256 | 256 | 128 | 128 | 64 | 64 | $3 \times 3$ | 1 | 1 |
| maxpool | 256 | 256 | 128 | 128 | 64 | 64 | $2 \times 2$ | 0 | 2 |
| conv5 | 512 | 512 | 256 | 256 | 128 | 128 | $3 \times 3$ | 1 | 1 |
| conv6 | 512 | 512 | 256 | 256 | 128 | 128 | $3 \times 3$ | 1 | 1 |
| maxpool | 512 | 512 | 256 | 256 | 128 | 128 | $2 \times 2$ | 0 | 2 |
| fc | 8192 | 10 | 4096 | 10 | 2048 | 10 | - | - | - |

