# OpenReview forum: "Prune or quantize? Strategy for Pareto-optimally low-cost and accurate CNN"
_ICLR.cc/2020/Conference — Reject_

### Official Review · AnonReviewer2 · 2019-10-22
**Official Blind Review #2**

**Rating:** 3

**Review:**

The paper proposes a new metric to evaluate both the amount of pruning and quantization. This metric is agnostic to the hardware architecture and is simply obtained by computing the Frobenius norm of some point-wise transformation of the quantized weights. They first show empirically that this Evaluation metric is correlated with the validation accuracy. Then use this metric to provide some general rules for pruning/quantizing to preserve the highest validation accuracy. Finally, they derive a strategy to perform pruning by monitory the signal to noise ratio during training and show experimentally that such method performs better than competing ones.

Pros: - Extensive experiments were performed to test the methods, and the results seem promising.

Cons: - The paper is not very clear, and the structure is somehow confusing.
- It is not easy at first to understand the experimental setup and requires to make a lot of guesses in my opinion.
- The paper didn't motivate properly the use of a hardware-agnostic metric in the context of the quantization and pruning. Isn't the ultimate goal of pruning/quantization is to optimize the run time/energy consumption of the specific device with the least compromise on the accuracy?

I feel that the paper currently jumps between very different ideas:
	- Evolution of the proposed metric during training: 2.3 and 2.5. While the 2.3, the take-home message is relatively clear:  the ESN is correlated with the validation accuracy, I don't fully get the point of section 2.5: It suggests that the optimizer does some sort of pruning just by choosing a higher learning rate.
	- Finding an optimal strategy for pruning/quantizing a network: 2.4 and 2.6. Those two sections are relatively clear, although I have some questions about the experiments.
	- Developing a new strategy based on the proposed metric to quantize and prune a network in a Pareto optimal sense: This is briefly and not very well explained in section 2.7, which sends back to 2.3, but it is hard to understand how it is exactly done. It seems that section 3 provides some empirical evidence supporting this strategy, but the description of the method is hidden in the experimental details.


Some questions:
- In figure 3, the blue dots represent validation vs ESN at each training iteration? What about the red plot, is it obtained by quantization of the parameters at different stages of training, or is it using the final parameters? Which equation was used to compute the red curve (2) or (4)?  How much quantization was performed? If the quantization was chosen to match the level of noise then it seems natural to expect such behavior in figure 3.

- In figure 4, how much pruning was performed for each network and was it the same quantization? In other words how each point in the plot was obtained?  The authors come to the conclusion that one should 'prune until the limit of the desired accuracy and then quantize', but it is hard for me to reach the same conclusions as I don't see the separate effect of pruning and quantization in those figures. Or maybe pruning is implicitly done by choosing a small network? In this case, it makes more sense, but still, some clarifications are needed.

- Which equation for the ESN was used to produce figure 5? Equation (2) or (4)?

- What is the Pareto frontier? I think it is worth first introducing this concept and describing more precisely how those curves are obtained. For someone who is not very familiar with these concepts, which is my case, it makes the reading very hard.

- How was the number of pruned filters computed in figure 5 (right)? I don't expect the solutions to be sparse during training, especially that no sparsity constraint was imposed, or was it?



-------------------------------------------------------------------------------
Revision:

Thank you for all the clarifications and the effort to make provide a clearer version of the paper.

Regarding section 2.7: ESNa FOR QUANTIZATION: Would it make sense to include the paragraph 2.7 at the end of 2.3, since it related to it and doesn't seem to require any of the intermediate subsections.

response to Comment 3: Unfortunately, I'm not convinced by the explanation about the effect of the lr on sparsity. The decay coefficient controls the saparsity indeed, but not the lr. That is because unlike the lr, the decay coefficient defines the cost functions to be optimized:  L+dc ||W||^2, while the lr corresponds simply to the discretization of some gradient flow.   For instance, in a deterministic and convex setting, the solution that is obtained would be the same, regardless of the chosen lr  ( provided the lr is smal enough so that the algorithm converges) see for instance [1].  In a non-convex and stochastic setting do the authors have a particular reference in mind? I'm not aware of such behavior. I would expect a similar sparsity if dc is kept fixed and only lr changed. Is it likely that with smaller lr, the algorithm just didn't have time to converge? This would explain why the obtained solutions were less sparse.
response to answer 5:  it is indeed well known that L1 norm induces sparsity, however l2 doesn't, it just encourages the weights to be smaller. In the optimization litterature sparsity of x% means x% of the parameters  are exactly 0. This is achieved with l1 norm, however l2 norm would only enforce that the coefficients are small but not necessarily 0 (see [1])
[1]:  ROBERT TIBSHIRANI, Regression Shrinkage and Selection via the Lasso.

Although the paper improved in terms of clarifications and experimental details, I still think it will benefit from additional work on careful interpretation of the results.












**Experience Assessment:**

I do not know much about this area.

**Review Assessment: Checking Correctness Of Derivations And Theory:**

N/A

**Review Assessment: Checking Correctness Of Experiments:**

I assessed the sensibility of the experiments.

**Review Assessment: Thoroughness In Paper Reading:**

I made a quick assessment of this paper.

---

> ### Author Response · Authors · 2019-11-14
> **Response to Reviewer 2 (1/3)**
>
> Thank you for detailed comments and questions. We reply to your comments and questions as follows.
>
> Comment 1
> >Cons: - The paper is not very clear, and the structure is somehow confusing.
> >- It is not easy at first to understand the experimental setup and requires to make a lot of guesses in my opinion.
>
> Thank you for kind feedbacks on our paper. We would add the information about experimental setup.
> We assume the main reasons why the structure is confusing are, as you pointed out,
> 1.	The message of section 2.5 is not clear.
> 2.	The discussions in section 2.7 (and section 2.3) interrupt the flow of the body contents of this paper.
> Let us explain some improvement plans below.
>
>
> Comment 2
> > - The paper didn't motivate properly the use of a hardware-agnostic metric in the context of the quantization and pruning. Isn't the ultimate goal of pruning/quantization is to optimize the run time/energy consumption of the specific device with the least compromise on the accuracy?
>
> Performances measured using existing hardware such as CPUs and GPUs cannot evaluate some cutting edge ideas proposed by researchers in algorithmic side, e.g. 1 or 2 bits quantization. For example, as discussed in section 2, the energy consumption of CPUs and GPUs does not decrease even when the bit width is reduced to 1 or 2 bits by quantization or when the number of non-zero weight parameters is reduced by pruning.
>
> Our motivation is to evaluate such ideas properly.
>
> Many researches of new hardware architecture handling the both quantization and pruning are ongoing [1], [2], [3], [4]. For instance, previous works [1], [2] proposed accelerators which support variable weight bit precision from 1 to 16 bit and achieve higher energy efficiency with lower bit precision by quantization.
> In addition, previous works [3], [4] proposed sparsity-aware accelerators which support sparse convolution and achieve higher energy efficiency with higher rate of zero-weights by pruning.
>
> Following the appearance of such dedicated accelerators, we expect that accelerators dedicated to both quantization and pruning will appear (or rather should be developed) in the near future.
>
> Our proposed metric allows us to estimate the performance when a quantized and pruned model runs on such devices that support both quantization and pruning.
>
> We agree that it is important to fine-tune models to optimize the computational cost of existing specific devices, and we should do that even after novel devices that support both quantization and pruning are developed.
>
> We believe, however, we also need such metric as our proposal for making sure that the both algorithmic and hardware architecture researches are on the same page for future emerging devices.
>
> References
> [1] A. Maki et al., “FPGA-based CNN processor with filter-wise-optimized bit precision”, In 2018 IEEE Asian Solid-State Circuits Conference (A-SSCC), pp. 47–50, Nov 2018. doi: 10.1109/ASSCC.2018.8579342.
> (We have referred this paper as “Maki et al., 2018” in the original manuscript.)
>
> [2] J. Lee et al., “UNPU: A 50.6TOPS/W Unified Deep Neural Network Accelerator with 1b-to-16b Fully-Variable Weight Bit-Precision”, In 2018 IEEE International Solid State Circuits Conference (ISSCC), pp. 218-220, Feb 2018. doi: 10.1109/ISSCC.2018.8310262.
>
> [3] Z. Yuan et al., “STICKER: A 0.41-62.1 TOPS/W 8bit Neural Network Processor with Multi-Sparsity Compatible Convolution Arrays and Online Tuning Acceleration for Fully Connected Layers”, In 2018 IEEE Symposium on VLSI Circuits, pp. 33-34, Jun 2018. doi: 10.1109/VLSIC.2018.8502404.
>
> [4] J. Song et al., “An 11.5TOPS/W 1024-MAC Butterfly Structure Dual-Core Sparsity-Aware Neural Processing Unit in 8nm Flagship Mobile SoC”, In 2019 IEEE International Solid State Circuits Conference (ISSCC), pp. 130-132, Feb 2019. doi: 10.1109/ISSCC.2019.8662476.
>
> # # # continued to next comment # # #

---

> > ### Author Response · Authors · 2019-11-14
> > **Response to Reviewer 2 (2/3)**
> >
> > # # # continued from previous comment (1/3) # # #
> >
> >  Comment 3
> > > - Evolution of the proposed metric during training: 2.3 and 2.5. While the 2.3, the take-home message is relatively clear: the ESN is correlated with the validation accuracy, I don't fully get the point of section 2.5: It suggests that the optimizer does some sort of pruning just by choosing a higher learning rate.
> >
> > Firstly, we explain the message of section 2.5. We apologize that the message of section 2.5 is unclear. In the experiment of section 2.5, we observed how weight parameters were optimized by SGD under low ESN condition. Based on the equation (4), following two conditions are possible under low ESN condition.
> >
> > (i)	A lot of small S/Ns are accumulated.      (e.g. S/N_i=0.1, i=1,2,,…,10)
> > (ii)	A few large S/Ns are accumulated. 	     (e.g. S/N_1=1, S/N_i=0, i=2,,…,10)
> >
> > Here, S/N represents signal-to-noise ratio. The condition of (i) corresponds to a fat model with a huge number of parameters quantized to extremely low bit width, and the condition of (ii) corresponds to a slim model with a small number of parameters quantized to moderately low bit width. Interestingly, when we see the right graph in figure 5 showing solutions found by the optimizer, the optimizer seems to give priority to reducing the number of parameters by pruning filters and increasing the amplitude of remaining parameters, which corresponds to the condition of (ii). This finding gave a hint for us to conclude that models with fewer parameters achieve far better accuracy in the region of low computational cost after quantization. We understand that to unveil the reason of this phenomenon is also necessary as future work.
> >
> > Secondly, we explain pruning filters by the optimizer, as a supplement.
> > In the experiment of section 2.5, we impose weight decay regularization on weights parameters during training. According to a previous work [5], the constraint such as L1 or L2 norm regularization has an effect of sparsity constraint on the weights. The effect of weight decay is determined by learning rate as well as decay coefficient, following the equation(*) of weight update.
> >  W^(t+1) = W^(t) – lr * {dL/dW^(t) + dc * W^(t)}	(*)
> > Here, lr represents learning rate, dc represents decay coefficient of weight decay, and L represents loss function. If the learning rate is higher, the decay effect will be larger and the effect of sparsity constraint will be also larger. As the result, in the right graph of figure 5, the higher the learning rate is, the larger the number of pruned filters is.
> >
> > Reference
> > [5] Y. Wang et al, “Nonstructured DNN weight pruning considered harmful”, CoRR, abs/1907.02124, 2019b. URL: http://arxiv.org/abs/1907.02124.
> > (We have referred this paper as “Wang et al., 2019b” in the first paragraph in section 2.6.)
> >
> >
> > Comment 4
> > > - Finding an optimal strategy for pruning/quantizing a network: 2.4 and 2.6. Those two sections are relatively clear, although I have some questions about the experiments.
> >
> > We apologize for the lack of experimental setup such as network architecture in section 2.4 and 2.6. We added the information as follows and updated the manuscript.
> >
> > Section 2.4
> > We added the information about network architectures of VGG7 in Appendix F, and modified the second to fifth sentences in the first paragraph as follows.
> > (Before): We evaluate 6 network architectures with different width and depth on CIFAR-10 dataset.
> > (After): We evaluate 6 network architectures with different depth and width: ResNet-8, 14, 20 and VGG7 with different channel width (1x, 0.5x, 0.25x) on CIFAR-10 dataset.
> > In this experiment, we employ networks with originally small number of parameters as slim models instead of pruning fat models.
> > The network architecture of ResNet is based on the original one without bottleneck architecture (He et al., 2016). The detailed network architecture of VGG7 is shown in Appendix F.
> >
> > Section 2.6
> > We modified the first sentence in the second paragraph as follows.
> > (Before): In this experiment, we take pretrained weight parameters and update them through SGD algorithm with ADMM regularization (Wang et al., 2019b) for a few additional epochs.
> > (After): In this experiment, we evaluate ResNet-20 on CIFAR-10 dataset. We take pretrained weight parameters and update them through SGD algorithm with ADMM regularization (Wang et al., 2019b) for a few additional epochs.
> >
> > # # # continued to next comment # # #

---

> > > ### Author Response · Authors · 2019-11-14
> > > **Response to Reviewer 2 (3/3)**
> > >
> > > # # # continued from previous comment (2/3) # # #
> > >
> > > Comment 5
> > > >- Developing a new strategy based on the proposed metric to quantize and prune a network in a Pareto optimal sense: This is briefly and not very well explained in section 2.7, which sends back to 2.3, but it is hard to understand how it is exactly done. It seems that section 3 provides some empirical evidence supporting this strategy, but the description of the method is hidden in the experimental details.
> > >
> > > Thank you for this comment. We agree that it should be explained in detail with more experimental results. In appendix B, we have shown a preliminary experimental result. We would appreciate it if you refer to that. Besides, we apologize for making you misunderstanding. The quantization policy mentioned in section 2.7 is not used for the experiment in section 3. Although section 2.7 is not the main body of our paper, we would like to share the interesting result showing one of the ESN-based potential applications. If you don’t think that section 2.7 may fit in the main topic of our paper, we consider moving the contents of section 2.7 to appendix. We would appreciate it if you could provide your comments or suggestions as one of our reviewers.
> > >
> > >
> > > Some questions:
> > > Question 1-(1)
> > > > In figure 3, the blue dots represent validation vs ESN at each training iteration? What about the red plot, is it obtained by quantization of the parameters at different stages of training, or is it using the final parameters? Which equation was used to compute the red curve (2) or (4)?
> > >
> > > Answer 1-(1)
> > > We are sorry for the lack of explanations. In figure 3, the blue dots represent validation vs ESN at each training iteration. The red plot is obtained by quantization of the final parameters. We used equation (2) to compute the red curves as we mentioned in the caption of figure 3. We expect the similar plot can be obtained by using equation (4) if quantization noise is approximated by uniform distribution (please refer to appendix E for details).
> > >
> > >
> > > Question 1-(2)
> > > > How much quantization was performed? If the quantization was chosen to match the level of noise then it seems natural to expect such behavior in figure 3.
> > >
> > > Answer 1-(2)
> > > The procedure to have figure 3 is as follows. We sweep the quantization step (\varDelta) as we define in the first paragraph of 4th page. When we set the quantization step to a value and quantize weight parameters, we can obtain the value of ESN_a by using equation (2). Then, we evaluate the validation accuracy of quantized model. We sweep the quantization step and plot the values of ESN_a and validation accuracy on the graph as red curves in figure 3. Interestingly, models during training and after quantization achieve similar validation accuracy at the same value of ESN_a, though the noise of each ESN_a is computed based on different concepts such as weight perturbation during SGD training and quantization noise.
> > >
> > >
> > > Question 2
> > > >- In figure 4, how much pruning was performed for each network and was it the same quantization? In other words how each point in the plot was obtained? The authors come to the conclusion that one should 'prune until the limit of the desired accuracy and then quantize', but it is hard for me to reach the same conclusions as I don't see the separate effect of pruning and quantization in those figures. Or maybe pruning is implicitly done by choosing a small network? In this case, it makes more sense, but still, some clarifications are needed.
> > >
> > > Answer 2
> > > We apologize for the lack of clarifications in this part. In section 2.4, as you guessed, we employ an originally small network as a slim model instead of pruning a fat model. We added the sentence to the updated manuscript in the first paragraph of section 2.4, as follows.
> > >
> > > “In this experiment, we employ networks with originally small number of parameters as slim models instead of pruning fat models.”
> > >
> > >
> > > Question 3
> > > >- Which equation for the ESN was used to produce figure 5? Equation (2) or (4)?
> > >
> > > Answer 3
> > > We are sorry for inadequate explanations. We used equation (4) and (5) to calculate ESN_a in figure 5. We added the sentence to the caption of figure 5 as follows, and uploaded the modified manuscript.
> > >
> > > “ESN_a is calculated by equation (4) and (5).”
> > >
> > > # # # continued to next # # #

---

> > > > ### Author Response · Authors · 2019-11-14
> > > > **Response to Reviewer 2 (4)**
> > > >
> > > > # # # continued from previous comment (3/3) # # #
> > > >
> > > > Question 4
> > > > > - What is the Pareto frontier? I think it is worth first introducing this concept and describing more precisely how those curves are obtained. For someone who is not very familiar with these concepts, which is my case, it makes the reading very hard.
> > > >
> > > > Answer 4
> > > > We appreciate your valuable advice regarding “Pareto frontier”. We give a supplementary explanation about that. In order to explain “Pareto frontier”, we describe multi-objective optimization problem. We assume the situation that we have two or more objective functions, and that we optimize the objective functions based on some criteria. Pareto optimality is a state that there is no alternative state that would make some objective functions better off without making any other objective functions worse off. Then, Pareto frontier is the set of Pareto optimal states for various conditions.
> > > >
> > > > For example, in the experiment of section 2.6, objective functions are ESN_a and validation accuracy. In the experiment, we evaluated only pruned, only quantized, and prune-then-quantize mixed models. At a certain value of ESN_a, the highest accuracy of them is picked up as the Pareto optimal point. In figure 6, the Pareto frontier is the plot of prune-then-quantize mixed model (red curve).
> > > >
> > > >
> > > > Question 5
> > > > >- How was the number of pruned filters computed in figure 5 (right)? I don't expect the solutions to be sparse during training, especially that no sparsity constraint was imposed, or was it?
> > > >
> > > > Answer 5
> > > > In the experiment of section 2.5, we impose L2 norm regularization (so-called weight decay) on weights parameters during training. According to a previous work [5], the constraint such as L1 or L2 regularization has an effect of sparsity constraint on the weights. Moreover, those regularization methods can be applied for pruning method, and we used this technique in section 2.6.

---

### Official Review · AnonReviewer1 · 2019-10-23
**Official Blind Review #1**

**Rating:** 3

**Review:**

This paper proposes an effective signal norm metric that measures the cost of the neural networks under both compute(ESN_a) or memory(ESN_d) in an ideal hardware setting. The authors then show that the slimmer models with fewer parameters are better than fatter models.

Strength

First of all, I like the discussion about different hardware(CPU) setups and corresponding cost. I also like the usage of pareto frontier to compare experimental methods. The proposed metric was simple but the author justifies it with respect to possible hardware setups in an ideal setting.

Weakness

The main drawback of the paper is the lack of novelty in proposed method. The metric itself appeared to be the main contribution, but as the author said, the metric was based on an ideal hardware setup that ignores the memory hierarchy and data transfer cost, which could be the bottleneck in reality.

More importantly, it is hard to use the empirical evaluation to justify the conclusion “fatter models are worse”. As we know that extremely low bit quantized models are very hard to train, and it is not surprising -- by setting the number of bits to extremely low bits, the models cannot recover in a few rounds of re-training.

This being said, I cannot think of a better alternative. While I do not think the experimental results offers new insights(other than it is hard to re-train lower bits models). How to quickly train low bits models remains an open question.

Finally given that there are already quite a few methods that prunes model based on real hardware evaluations, it would be great to compare to these methods as well.

Because the above weakness of the paper, I think this paper should not be accepted to the program.



**Experience Assessment:**

I have read many papers in this area.

**Review Assessment: Checking Correctness Of Derivations And Theory:**

I assessed the sensibility of the derivations and theory.

**Review Assessment: Checking Correctness Of Experiments:**

I carefully checked the experiments.

**Review Assessment: Thoroughness In Paper Reading:**

I read the paper thoroughly.

---

> ### Author Response · Authors · 2019-11-15
> **Response to Reviewer #1 (1/2)**
>
> We appreciate your detailed comments and insightful feedback on our paper. We reply to your comments as follows.
>
> Comment 1
> >The main drawback of the paper is the lack of novelty in proposed method. The metric itself appeared to be the main contribution, but as the author said, the metric was based on an ideal hardware setup that ignores the memory hierarchy and data transfer cost, which could be the bottleneck in reality.
>
> Answer 1
> We mainly focus on CNN, which has large computational intensity and therefore is less prone to be memory bottleneck. Although pruning and quantization are typical approach to reduce the computational cost of CNN, existing hardware such as CPU and GPU cannot exploit them, and e.g. a metric of speed measured using CPU cannot evaluate the pruned and quantized models properly.
>
> Our motivation is to provide a metric which can evaluate them.
> (We would like you to see the response to Comment 2 in the discussion with Reviewer #2 as well.)
> So the proposed metric of ESN has such novelty that is able to evaluate pruned and quantized models.
>
> Although the proposed metric assumes ideal hardware, we believe that our assumption is reasonable.
> For example, there have been a lot of proposals that handle either quantized or pruned CNN models efficiently [1-4], and the fact that they demonstrated the energy efficiency was improved by applying quantization or pruning supports our assumption.
> (We would like you to see the response to Negative Points 1 & 2 in the discussion in Reviewer #4.)
>
> We agree that even in CNN or a situation with high computational intensity, the energy consumption for memory access is still significant. Pruning and quantization, which reduces the amount of data transferred between memory and processor, contributes to reducing memory accesses. The tendency to reduce memory accesses by pruning and quantization is similar to that to reduce computational costs. Hence, ESN allows us to estimate the tendency to reduce memory accesses as well as computational costs by pruning and quantization.
>
> We agree that it is important to fine-tune models to optimize the computational cost considering memory hierarchy and data transfer cost, and we should do that even after novel devices that support both quantization and pruning are developed. We believe, however, we also need such metric as our proposal for making sure that the both algorithmic and hardware architecture researches are on the same page for future emerging devices.
>
>
> Comment 2
> > More importantly, it is hard to use the empirical evaluation to justify the conclusion “fatter models are worse”. As we know that extremely low bit quantized models are very hard to train, and it is not surprising -- by setting the number of bits to extremely low bits, the models cannot recover in a few rounds of re-training.
> This being said, I cannot think of a better alternative. While I do not think the experimental results offers new insights (other than it is hard to re-train lower bits models). How to quickly train low bits models remains an open question.
>
> Answer 2
> We are sorry for making you misunderstanding. The conclusion of our paper is not “fatter models are worse” but “the accuracy of fat models quantized to extremely low bit is worse than that of quantized slim models in the region of low computational cost.”
> We assume a use-case scenario where a user (e.g. at edge side) quantizes pretrained models in order to obtain models which can maintain original accuracy with a low computational cost. In that case, it is not necessary to train low-bit models through backpropagation. Therefore, the situation that the user trains extremely low-bit-quantized models, which is very hard as you pointed out, does not occur. The user only fine-tunes models quantized to low bits for tuning statistic parameters in BatchNorm layers. Then, the user can tune the parameters quickly without backpropagation or any labeled data [5].
> For the above reason, to re-train low bit models and to train quickly low-bit models are outside the scope of this study.
>
> # # # continued to next comment # # #

---

> > ### Author Response · Authors · 2019-11-15
> > **Response to Reviewer #1 (2/2)**
> >
> > # # # continued from previous comment (1/2) # # #
> >
> > Comment 3
> > > Finally given that there are already quite a few methods that prunes model based on real hardware evaluations, it would be great to compare to these methods as well.
> >
> > Answer 3
> > Certainly, there are a lot of conventional methods that prune model based on evaluations with existing hardware. However, our goal is to properly evaluate models where both pruning and quantization are applied, and as we described above, the existing hardware cannot reflect the effect of quantization in addition to that of pruning. Therefore, we think we should evaluate and compare methods for reducing computational costs of CNN including our proposal and also conventional pruning and/or quantization schemes using hardware that supports both pruning and quantization. We believe such hardware will appear in the near future. Although we used weight-regularization based pruning method in this study, any pruning method can be employed to produce a pruned slim model. We expect that such hardware can fully exploit the benefit of quantization, taking advantage of the best pruning method.
> >
> >
> > References
> > [1] A. Maki et al., “FPGA-based CNN processor with filter-wise-optimized bit precision”, In 2018 IEEE Asian Solid-State Circuits Conference (A-SSCC), pp. 47–50, Nov 2018. doi: 10.1109/ASSCC.2018.8579342.
> > (We have referred this paper as (Maki et al., 2018) in the original manuscript.)
> >
> > [2] J. Lee et al., “UNPU: A 50.6TOPS/W Unified Deep Neural Network Accelerator with 1b-to-16b Fully-Variable Weight Bit-Precision”, In 2018 IEEE International Solid State Circuits Conference (ISSCC), pp. 218-220, Feb 2018. doi: 10.1109/ISSCC.2018.8310262.
> >
> > [3] Z. Yuan et al., “STICKER: A 0.41-62.1 TOPS/W 8bit Neural Network Processor with Multi-Sparsity Compatible Convolution Arrays and Online Tuning Acceleration for Fully Connected Layers”, In 2018 IEEE Symposium on VLSI Circuits, pp. 33-34, Jun 2018. doi: 10.1109/VLSIC.2018.8502404.
> >
> > [4] J. Song et al., “An 11.5TOPS/W 1024-MAC Butterfly Structure Dual-Core Sparsity-Aware Neural Processing Unit in 8nm Flagship Mobile SoC”, In 2019 IEEE International Solid State Circuits Conference (ISSCC), pp. 130-132, Feb 2019. doi: 10.1109/ISSCC.2019.8662476.
> >
> > [5] S. Sasaki, et al., “Post training weight compression with distribution-based filter-wise quantization step”, In 2019 IEEE Symposium in Low-Power and High-Speed Chips (COOLCHIPS), pp. 1–3, April 2019. doi: 10.1109/CoolChips.2019.8721356.
> > (We have referred this paper as “Sasaki et al., 2019” in the original manuscript)

---

### Official Review · AnonReviewer4 · 2019-11-02
**Official Blind Review #4**

**Rating:** 3

**Review:**

The authors propose a hardware-agnostic metric called effective signal norm (ESN) to measure the computational cost of convolutional neural networks. This metric aims to fairly measure the effects of pruning and quantization. What’s more, based on the metric, the authors demonstrate that models with fewer parameters achieve far better accuracy after quantization. A large number of experiments are carried out to prove the effects of the metric and related conclusions, however, several experiments and arguments are confusing.

Please see my detailed comments below.

Positive Points:
1. The authors propose a hardware-agnostic metric named effective signal norm (ESN), aiming to measure the computational cost of the pruned and quantized models.

2. Based on the proposed metric, the authors conclude that a moderately quantized slim model with fewer weight parameters achieves better performance rather than an extremely quantized fat model with huge number of parameters.

3. This paper presents many interesting assumptions and possibilities, which can be further researched and explored in the future.


Negative Points:
1. The authors argue that ESN is a hardware-agnostic metric. However, ESN is based on ideal hardware, where the energy consumption is linearly proportional to the number of non-zero weight parameters and monotonically depends on the bit-width of weight parameters. Therefore, ESN is not suitable for existing hardware.

2. Both of ESN_a and ESN_d are based on the assumptions instead of the fact, which is not convincing. What’s more, the assumptions are hard to be proved.

3. The experiments are not strong enough to support the author's conclusions. For example, the authors argue that “models with fewer parameters achieve far better accuracy in low computational cost region after quantization”. However, this conclusion is only based on the ESN metric. Since ESN is based on some assumptions, this conclusion is not convincing.

4. Please give more details about the quantitative relation between the quantization noise and the perturbation of weight parameters during SGD training.

5. In terms of Figure 2, the author's description is not objective. The similarity between the ESN_a curves and the validation accuracy curves is not very high. Besides, the authors mention that “after steep rise at 80 epoch, both ESN_a and validation accuracy decrease gradually”. However, the performance degradation in Figure 2 is not obvious. Therefore, the conclusions drew by the observation are also unreliable.

6. The ESN can not make an accurate evaluation of a model. The trade-off between accuracy, speed, model size is often required in model compression. However, the ESN can not make the trade-off between them. Moreover, the ESN does not have a clear application scenario.

Minor issues:
1.	There are lots of spelling and grammar mistakes in the paper, such as “an ideal hardware ”, “a extremely quantized fat model” and so on.


**Experience Assessment:**

I have published one or two papers in this area.

**Review Assessment: Checking Correctness Of Derivations And Theory:**

I carefully checked the derivations and theory.

**Review Assessment: Checking Correctness Of Experiments:**

I carefully checked the experiments.

**Review Assessment: Thoroughness In Paper Reading:**

I read the paper thoroughly.

---

> ### Author Response · Authors · 2019-11-12
> **Response to Reviewer 4 (1/2)**
>
> Thank you for detailed comments and insightful feedback on our paper.
> In the following, we reply to your comments, especially relating to negative points.
>
> > Negative Points:
> > 1. The authors argue that ESN is a hardware-agnostic metric. However, ESN is based on ideal hardware, where the energy consumption is linearly proportional to the number of non-zero weight parameters and　monotonically depends on the bit-width of weight parameters. Therefore, ESN is not suitable for existing hardware.
> > 2. Both of ESN_a and ESN_d are based on the assumptions instead of the fact, which is not convincing. What’s more, the assumptions are hard to be proved.
> > 3. The experiments are not strong enough to support the author's conclusions. For example, the authors argue that “models with fewer parameters achieve far better accuracy in low computational cost region after quantization”. However, this conclusion is only based on the ESN metric. Since ESN is based on some assumptions, this conclusion is not convincing
> > 6. The ESN cannot make an accurate evaluation of a model. The trade-off between accuracy, speed, model size is often required in model compression. However, the ESN cannot make the trade-off between them.
>
> In the beginning, we reply to 1st, 2nd, 3rd and 6th negative comments together.
>
> Although 1st to 3rd negative comments are based on a skeptical view about adopting a "virtual" metric of ESN to evaluate models instead of performances such as speed measured using existing hardware, 6th negative comment is exactly relevant to the motivation for adopting the “virtual” metric of ESN.
>
> For example, as discussed in section 2, the energy consumption of CPUs or GPUs, which are the most widely utilized existing hardware for general purposes, does not decrease even when the bit width is reduced to 1 or 2 bits by quantization or when the number of non-zero weight parameters is reduced by pruning. In other words, they cannot fully exploit the benefits of quantization and pruning.
>
> It is clear that we should not use performances such as energy consumption and speed for evaluating models where quantization or pruning is applied. One reason is that such metric evaluated with existing hardware cannot make the trade-off between accuracy, speed, model size (this is the challenge as you mentioned in the 6th comment).
>
> Actually, a lot of hardware architecture dedicated to either quantization or pruning have been proposed recently. For example, previous works [1], [2] proposed accelerators which support variable weight bit precision from 1 to 16 bit and achieve higher energy efficiency with lower bit precision by quantization. The previous work [2] shows an experimental result that the energy efficiency is improved from 3.08 to 11.6 [TOPS/W] when the bit width is reduced from 16 to 4 bit by quantization.
> In addition, previous works [3], [4] proposed sparsity-aware accelerators which support sparse convolution and achieve higher energy efficiency with higher rate of zero-weights by pruning. The previous work [4] shows the energy efficiency is improved from 0.41 to 62.1 [TOPS/W] when the sparse rate of weights and activations is changed from 5 to 95 %.
>
> The assumption for ESN is derived from these facts. Also, following the appearance of such dedicated accelerators, we expect that accelerators dedicated to both quantization and pruning will appear (or rather should be developed) in the near future, and that those accelerators can satisfy our assumption.
>
> As for 1st comment, it is true that "ESN is not suitable for existing hardware", because our target is not existing hardware but more properly designed hardware that should appear in the future.
>
> As for 2nd comment, we believe our assumption that the energy consumption is linearly proportional to the number of non-zero weight parameters and monotonically depends on the bit-width of weight parameter is reasonable. There are many supporting works such as [1], [2], [3] and [4], which have shown experimental results that the energy consumption is reduced and energy efficiency is improved if the data size for model parameters is reduced by quantization or pruning.
>
> As for 3rd comment, since the assumption is based on those facts, the conclusion based on the evaluation using ESN should be convincing.
>
> # # continued to next comment # #

---

> > ### Author Response · Authors · 2019-11-12
> > **Response to Reviewer 4 (2/2)**
> >
> > # # continued from previous comment (1/2) # #
> >
> > As for the trade-off in 6th comment, we think ESN can make the trade-off between model size, accuracy and execution time or speed in model compression (detailed explanations to the reason is as follows[*]). On the other hand, the metric evaluated with existing hardware (e.g. execution time on CPU) cannot make such trade-off, because existing hardware cannot fully exploit the benefits of quantization and pruning.
> >
> > [*] Detailed explanations to the trade-off, mentioned in 6th comment
> > In our paper, the assumption for ESN definition is related to energy consumption that is consumed when the model is executed on the ideal hardware. Then, if ESN is reduced by pruning or quantization, the model size and energy consumption will decrease, and the accuracy will deteriorate (as you can see the tendency in figure 2, 3, and 4), and vice versa. Therefore, ESN can make the trade-off between model size, energy consumption and accuracy.
> > Here, the assumption for ESN definition can be also related to execution time that the ideal hardware takes when the model is executed on the hardware. Then, ESN can make the trade-off between model size, accuracy and execution time (or speed), as is the case with the definition based on energy consumption.
> >
> > As for clear applications of ENS in 6th comment, we applied ESN to evaluate models where both quantization and pruning were applied, in section 3. In addition, we expect ESN can be also applied to evaluate computational cost of accelerators dedicated to both quantization and pruning, which we believe will appear in the near future. Besides, we think evaluations by using ESN can be applied to determining quantization policy (as we mentioned in section 2.7 and appendix B). Detailed experiments and considerations are left for future work.
> >
> > References
> > [1] A. Maki et al., “FPGA-based CNN processor with filter-wise-optimized bit precision”, In 2018 IEEE Asian Solid-State Circuits Conference (A-SSCC), pp. 47–50, Nov 2018. doi: 10.1109/ASSCC.2018.8579342.
> > (We have referred this paper as (Maki et al., 2018) in the original manuscript.)
> >
> > [2] J. Lee et al., “UNPU: A 50.6TOPS/W Unified Deep Neural Network Accelerator with 1b-to-16b Fully-Variable Weight Bit-Precision”, In 2018 IEEE International Solid State Circuits Conference (ISSCC), pp. 218-220, Feb 2018. doi: 10.1109/ISSCC.2018.8310262.
> >
> > [3] Z. Yuan et al., “STICKER: A 0.41-62.1 TOPS/W 8bit Neural Network Processor with Multi-Sparsity Compatible Convolution Arrays and Online Tuning Acceleration for Fully Connected Layers”, In 2018 IEEE Symposium on VLSI Circuits, pp. 33-34, Jun 2018. doi: 10.1109/VLSIC.2018.8502404.
> >
> > [4] J. Song et al., “An 11.5TOPS/W 1024-MAC Butterfly Structure Dual-Core Sparsity-Aware Neural Processing Unit in 8nm Flagship Mobile SoC”, In 2019 IEEE International Solid State Circuits Conference (ISSCC), pp. 130-132, Feb 2019. doi: 10.1109/ISSCC.2019.8662476.
> >
> >
> > Here, we reply to 4th, 5th comments and minor issues.
> >
> > > 4. Please give more details about the quantitative relation between the quantization noise and the perturbation of weight parameters during SGD training.
> >
> > We are sorry that this part was inadequate in the original manuscript. We agree that we should show more detailed experimental results about the “quantitative” relation between the quantization noise and the perturbation of weight parameters during SGD training. If necessary, we would modify the word of “quantitative” to more appropriate description, or delete it.
> >
> > > 5. In terms of Figure 2, the author's description is not objective. The similarity between the ESN_a curves and the validation accuracy curves is not very high. Besides, the authors mention that “after steep rise at 80 epoch, both ESN_a and validation accuracy decrease gradually”. However, the performance degradation in Figure 2 is not obvious. Therefore, the conclusions drew by the observation are also unreliable.
> >
> > We apologize for unclear graphs in figure 2 about validation accuracy and ESN_a. We revised the graphs as follows, and updated the manuscript.
> > ・ The range of vertical axis is changed to be narrow.
> > ・ Moving average curves between 3-epochs are added over the original plots.
> > * The data of original plots is not changed.
> >
> > > Minor issues:
> > > 1. There are lots of spelling and grammar mistakes in the paper, such as “an ideal hardware ”, “a extremely quantized fat model” and so on.
> >
> > According to minor issues such as spelling and grammar mistakes in our paper, we will have it proofread by native speakers and modify them before uploading again the revised manuscript by the end of discussion term (15th Nov.).

---

### Author Response · Authors · 2019-11-12
**Upload the revised manuscript**

Thank all reviewers for insightful comments and feedback on our paper.
We uploaded the revised manuscript.
We revised the manuscript as follows.

# We modified right graph in figure 2 and the caption as follows.
・ The range of vertical axis is changed to be narrow.
・ Moving average curves between 3-epochs are added over the original plots.
* The data of original plots is “not changed”.
・ We added the sentence: In the right graph, moving average curve between 3-epochs is overlapped on each plot, to the caption.

# We added the information about network architectures in section 2.4 and 2.6.
- the forth sentence in the first paragraph of Section 2.4
- the first sentence in the second paragraph of Section 2.6
# In the caption of figure 5, we added the explanation about calculation of ESN_a.
# We added Appendix F for the more detailed information about VGG7.

---

> ### Author Response · Authors · 2019-11-14
> **Upload the revised manuscript 2**
>
> We uploaded the revised manuscript. We revised the manuscript as follows.
>
> # We added the following sentence about experiments to the first paragraph in Section 2.4, taking the feedbacks from reviewer #2 as reference.
>
> "In this experiment, we employ networks with originally small number of parameters as slim models instead of pruning fat models."

---

> > ### Author Response · Authors · 2019-11-15
> > **Upload the revised manuscript 3**
> >
> > We had the manuscript proofread by native speakers and modified spelling and grammar mistakes. We uploaded the revised manuscript.

---

### Decision · Program_Chairs · 2019-12-19

**Decision:**

Reject

**Comment:**

The authors propose a hardware-agnostic metric called effective signal norm (ESN) to measure the computational cost of convolutional neural networks. They then demonstrate that models with fewer parameters achieve far better accuracy after quantization. The main novelty is on the metric ESN. However, ESN is based on ideal hardware, and thus not suitable for existing hardware. Assumptions made in the paper are hard to be proved. Experimental results are not convincing, and related pruning methods are not compared. Finally, the paper is not written clearly, and the structure and some arguments are confusing.